# Measurement Report: Rapid decline of aerosol absorption
# coefficient and aerosol optical properties effects on radiative
# forcing in an urban area of Beijing from 2018 to 2021
Xinyao Hu[1,2], Junying Sun[1,3*], Can Xia[1,4], Xiaojing Shen[1], Yangmei Zhang[1], Quan Liu[1],
Zhaodong Liu[1,4], Sinan Zhang[1], Jialing Wang[1], Aoyuan Yu[1,2], Jiayuan Lu[1], Shuo Liu[1], and
Xiaoye Zhang[1]
[1]State Key Laboratory of Severe Weather & Key Laboratory of Atmospheric Chemistry of
CMA, Chinese Academy of Meteorological Sciences, Beijing 100081, China
[2]University of Chinese Academy of Sciences, Beijing 100049, China
[3]State Key Laboratory of Cryospheric Science, Northwest Institute of Eco-Environment
and Resources, Chinese Academy of Sciences, Lanzhou 730000, China
[4]Nanjing University of Information Science & Technology, Nanjing 210044, China
*Correspondence to: Junying Sun (jysun@cma.gov.cn)
**Abstract**
Reliable observations of aerosol optical properties are crucial for quantifying the
radiative forcing of climate. The simultaneous measurements of aerosol optical properties
at three wavelengths for $PM_1$ and $PM_{10}$ were conducted in urban Beijing from March 2018
to February 2022. The aerosol absorption coefficient ($\sigma_{ab}$) at 550 nm of $PM_{10}$ and $PM_1$
decreased by 55.0% and 53.5% from 2018 to 2021. Significant reduction in $\sigma_{ab}$ may be
related to reduced primary emissions caused by effective air pollution control measures.
$PM_{2.5}$ mass concentration decreased by 34.4% from 2018 to 2021. SSA increased from
$0.89 \pm 0.04$ for $PM_{10}$ ($0.87 \pm 0.05$ for $PM_1$) in 2018 to $0.93 \pm 0.03$ for $PM_{10}$ ($0.91 \pm 0.04$
for $PM_1$) in 2021. Increasing SSA and decreasing $PM_{2.5}$ mass concentration suggest that
the fraction of absorbing aerosols decreased with improved air quality due to pollution
control measure-taking. The annual average submicron absorption ratio (Rab) increased
from 86.1% in 2018 to 89.2% in 2021, suggesting that fine particles are the main
contributors to total $PM_{10}$ absorption and that the fine particles to absorption became more
important. Absorption Angstrom exponent (AAE) in winter decreased from 2018 to 2021,
implying a decreasing contribution from brown carbon to light absorption, which may
relate to the reduced emissions of biomass burning and coal combustion. During the study
period, aerosol radiative forcing efficiency became more negative mainly influenced by
increasing SSA, and was $-27.0$ and $-26.2$ W m$^{-2}$ AOD$^{-1}$ for $PM_{10}$ and $PM_1$ in 2021. Higher
$\sigma_{ab}$ and $PM_{2.5}$ mass concentrations were primarily distributed in clusters 4 and 5,
transported from the south and the west of Beijing each year. $\sigma_{ab}$ and $PM_{2.5}$ corresponding
to clusters 4 and 5 decreased evidently from 2018 to 2021, which may result from the
control of source emissions in surrounding regions of Beijing. The 4-year data presented
in this study provide critical optical parameters for radiative forcing assessment within two
size ranges and are helpful for evaluating the effectiveness of clean air action.
**1 Introduction**
Atmospheric aerosols perturb the Earth's atmospheric radiation balance and climate
forcing by directly affecting the scattering and absorption of solar radiation (Charlson et
al., 1992; Jacobson, 2001) but also indirectly affecting cloud reflectivity and precipitation
processes (Twomey, 2007). Light-scattering aerosols contribute to offsetting the warming
effect of $CO_2$, while absorbing aerosols contribute to the heating of the atmosphere (Bond
and Bergstrom, 2007), and produce a positive radiative forcing (Segura et al., 2016). The
largest contribution to aerosol absorption is from black carbon (BC), which absorbs
strongly over the entire solar spectrum (Bond and Bergstrom, 2007). Dust and brown
carbon (BrC) are also light absorption aerosols, which strongly absorb in the ultraviolet
(UV) spectrum. Globally, aerosols contributed an effective radiative forcing (ERF) of –1.3
$\pm$ 0.7 W m$^{-2}$, and the ERF due to emissions of BC is now estimated to be 0.11 (–0.20 to
0.42) W/m$^2$ between 1750 to 2019 (Szopa et al., 2021). However, aerosol properties are
highly spatial and temporal variable, which results in radiative forcing variation from local
to global scales and creates an observational challenge (Collaud Coen et al., 2013; Ealo et
al., 2018; Andrews et al., 2011). Therefore, reliable observations of aerosol optical
properties are crucial for quantifying the radiative forcing of climate.
In order to assess the role of aerosols on climate forcing accurately, a set of parameters
that describe aerosol's optical properties are needed, such as scattering coefficient ($\sigma_{sp}$),
absorption coefficient($\sigma_{ab}$), backscatter fraction(b) and single scattering albedo (SSA). SSA
is a key variable that determines the magnitude and the sign of the aerosol forcing (J.
Hansen et al., 1997; Lee et al., 2007; Li et al., 2022a; Zhang et al., 2020). Previous studies
found that SSA values range from slightly less than 0.8 to almost purely scattering particles
with SSA close to 1 at worldwide locations (Laj et al., 2020; Pandolfi et al., 2018), and
higher SSA values indicate a tendency towards a cooling effect (Li et al., 2022a). The
backscatter fraction (b) describes how much aerosol particles scatter radiation in the
backward hemisphere compared with the total scattering, which is a crucial variable for
aerosol radiative forcing efficiency (RFE) calculations (Andrews et al., 2011; Sheridan and
Ogren, 1999; Luoma et al., 2019). Previous studies found that the magnitude of RFE
increases with increasing b (Shen et al., 2018). Typical values of b for the atmospheric
aerosol at 550 nm were from approximately 0.05 to 0.20 (Titos et al., 2021).
Besides, aerosol optical properties are wavelength-dependent, absorption Angstrom
exponent (AAE) describes the spectral dependence of light absorption by aerosols and is
typically used to differentiate between different aerosol types (Helin et al., 2021). The AAE
for fresh BC is ~1, indicating "weak" spectral dependence of light absorption (Bond et al.,
2013; Bond and Bergstrom, 2007), and the AAE >1 indicates the presence of BrC or dust,
which tend to exhibit absorption that increases sharply as wavelength decreases
(Moosmüller et al., 2009; Lack and Cappa, 2010). Thus, obtaining the aerosol absorption
coefficient at different wavelengths is essential and can be helpful to differentiate between
different aerosol types.
As one of the world's most populous and rapidly developing megacities, Beijing
experienced rapid economic growth and urbanization, accompanied by severe air pollution.
Many in-situ measurements of aerosol optical properties have been conducted in Beijing
(Bergin et al., 2001; He et al., 2009; Garland et al., 2009; Jing et al., 2015; Wang et al.,
2019; Zhao et al., 2019; Xia et al., 2020). Previous studies found that high aerosol loading
leads to large $\sigma_{ab}$ in Beijing (Jing et al., 2015; Garland et al., 2009; Bergin et al., 2001).
Moreover, the AAE showed significant seasonal variations in Beijing. Significantly higher
AAE in winter than in summer highlights the important role of absorption of non-BC
components (e.g. BrC) in winter (Xie et al., 2020; Xia et al., 2020). In order to reduce
emissions and improve air quality, the government implemented strict pollution control
measures (Xu and Zhang, 2020). Significant decreases in $PM_{2.5}$ mass concentrations were
found in Beijing and the annual mean elemental carbon (EC) concentrations declined from
4.0 to 2.6 µg m$^{-3}$ from March 2013 to February 2018 in Beijing (Ji et al., 2019). Xia et al.
(2020) separated and quantified the effects of emission control and meteorological
transport variability on BC loading from 2015 to 2017 in north China Plain. However, the
environmental effects caused by emission controls are related to not only their mass
concentrations, but also their optical properties and radiative effect (Luo et al., 2020).
Therefore, it's necessary to investigate the multiple-year variations in aerosol optical
properties and radiative effect in providing a comprehensive understanding of the effects
of emission control. Wang et al. (2019) found that absorption coefficient ($\sigma_{ap}$) for $PM_{2.5}$
decreased from 2014 to 2017, with a significant decrease of $\sigma_{ap}$ in autumn. Sun et al. (2022)
estimated that the direct radiative forcing of BC decreased by 67% from +3.36Wm$^{-2}$ in
2012 to +1.09 Wm$^{-2}$ in 2020. However, these studies were mostly conducted with
conventional total suspended particulate (TSP) cyclone, $PM_{2.5}$ size cut, or $PM_{10}$ size cut.
Few studies focused on the sub-micron and super-micron particle optical properties and
estimated aerosol radiative effect in the post-"Action Plan on Prevention and Control of
Air Pollution" era. Acquiring the aerosol optical for the total (< 10µm diameter) and
submicron aerosol is also in line with the aerosol advisory group of the Global Atmosphere
Watch recommendation (WMO/GAW, 2016).
In this study, the simultaneous measurements of aerosol optical properties at three
wavelengths for $PM_1$ and $PM_{10}$ were conducted in urban Beijing from March 2018 to
February 2022. The annual, seasonal, and diurnal variations of aerosol optical properties
for two size cuts were investigated. The scattering properties of aerosols for two size ranges
(PM$_{10}$ and PM$_1$) under dry conditions observed in Beijing have been analyzed in detail by
Hu et al. (2021). Thus, this study mainly focused on the variation of aerosol absorption
coefficient, single scattering albedo, and absorption Angstrom Exponent for PM$_{10}$ and PM$_1$.
Moreover, the aerosol radiative effects in two size cuts were estimated. Finally, the
transport and its impact on aerosol optical properties were analyzed. The 4-year data
presented in this study provide key optical parameters for radiative forcing assessment
within two size ranges and are helpful for evaluating the effectiveness of clean air action.
**2 Instrumentation and methods**
**2.1 Site description**
The sampling site in this study is located on the roof of the Chinese Academy of
Meteorological Sciences (CAMS, 116°19′ E, 39°57′ N, 46 m a.s.l) in Beijing, which is a
typical urban site in the northwest of Beijing between the 2nd and 3rd ring roads. The
laboratory is on the roof of CAMS building, and the measurements are taken at 53 m above
ground level. The site is mainly influenced by local emissions from residential living and
traffic pollution (Xia et al., 2019).
**2.2 Instruments and measurements**
The ambient air was sampled into a PM$_{10}$ impactor with 16.7 LPM and then to an
adsorption aerosol dryer, which controlled the relative humidity (RH) of sample air below
30% (Tuch et al., 2009). The dried aerosol sample passes through switched impactors that
toggle the aerosol size cut between 1.0 µm (<1 µm) and 10 µm (<10 µm) aerodynamic
particle diameters every 30 min, thus allowing to measure both fine and coarse particles

(Hu et al., 2021). The sample aerosol was then passed into the Nephelometer (TSI Inc., Model 3563) and Tricolor Absorption Photometer (TAP, Brechtel Manufacturing, Inc., Hayward, CA, USA).

The integrating nephelometer measured the scattering coefficient ($\sigma_{sp}$) (angular range of 7–170°) and backscattering coefficient ($\sigma_{bsp}$) (angular range of 90–170°) at 450, 550, and 700 nm. The scattering and backscattering coefficient were corrected for truncation and instrument non-idealities using the method described by Anderson and Ogren (1998). Details are given in Hu et al. (2021). To ensure the data's accuracy and reliability, the nephelometer was calibrated regularly using filtered ambient air using a HEPA filter and $CO_2$ with a purity of 99.999%. A zero-check was automatically performed once per hour to obtain a nephelometer background.

TAP  measures absorption coefficient ($\sigma_{ab}$) at 465, 520, and 640 nm with the 47 mm diameter, glass-fiber filter and is a commercially available version of the continuous light absorption photometer (CLAP), which is low cost and high sensitivity (Ogren et al., 2017). The TAP comprises eight sample spots and two reference spots. The aerosol-laden air passes through one sample spot at a time, which allows for 8 times the filter lifetime compared to single-spot photometers (Davies et al., 2019). Unlike the Multi-Angle Absorption Photometer (MAAP), TAP does require a co-located aerosol light scattering or extinction measurement to derive aerosol light absorption (Ogren et al., 2017). Thus, simultaneous observation of aerosol light scattering has been measured and used to correct absorption data. When the Nephelometer and TAP were calibrated or malfunctioning, no data are available. During this study, 84% of the data was effective.

## 2.3. Data processing

The TAP measures the light transmitted through a filter as particles are deposited onto the filter. The filter attenuation coefficient ($\sigma_{atn}$), at a specific wavelength ($\lambda$), can be determined as:

$$\sigma_{atn}(\lambda) = \frac{A}{Q} \times \frac{\Delta atn(\lambda)}{\Delta t} \tag{1}$$

where $\Delta atn\ (\lambda)$ is the filter attenuation at times t1 and t2, A is the area of on the filter, and Q is the sample flow rate through the filter.

In order to correct the error caused by multiple scattering and filter loading, the aerosol light absorption coefficient ($\sigma_{ab}\ (\lambda)$) was corrected based on the methods of Bond et al. (1999) and Ogren et al. (2017). First, the effect of filter loading was calibrated based on Eq. (2):

$$\sigma_{ab}(\lambda)\_raw = \frac{0.85 \times \sigma_{atn}(\lambda)}{K_2 \times (1.0796 \times Tr(\lambda) + 0.71)} \tag{2}$$

Then, $\sigma_{ab}\ (\lambda)\_raw$ at 465, 520, and 640 nm were adjusted to the wavelength of the light scattering coefficient based on the calculated AAE. Finally, the multiple scattering effect was corrected based on Eq. (3):

$$\sigma_{ab}(\lambda) = \sigma_{ab}(\lambda)\_raw - \frac{K_1 \times \sigma_{sp}(\lambda)}{K_2} \tag{3}$$

where $Tr\ (\lambda)$ is the normalized filter transmittance at time t relative to transmittance at the start of sampling (t =0) and $\sigma_{sp}$ is the aerosol light-scattering coefficient at 450, 550, and 700 nm measured by the nephlometer. $K_1$ and $K_2$ were derived by Bond et al. (1999) as $K_1$ =0.02±0.02 and $K_2$ =1.22±0.20, where the uncertainties are given for the 95% confidence level.

Using the corrected absorption coefficient data, the following parameters were
calculated:
Absorption Angstrom exponent (AAE) describes the spectral dependence of light
absorption.
$AAE = -\dfrac{\ln(\sigma_{ab}^{\lambda 1}/\sigma_{ab}^{\lambda 2})}{\ln(\lambda 1/\lambda 2)}$           (4)
The submicron absorption ratio (Rab) is determined as the ratio of the absorption
coefficients for $PM_1$ and $PM_{10}$.
$Rab = \dfrac{\sigma_{ab}(D<1\mu m)}{\sigma_{ab}(D<10\mu m)}$           (5)
where $\sigma_{ab}$ (D<1 μm) and $\sigma_{ab}$ (D<10 μm) are $\sigma_{ab}$ for particle diameters <1 μm and 10
μm, respectively.
Aerosol radiative forcing efficiency (RFE) at top-of-the-atmosphere (TOA) is a
simplified formula that describes how large of an impact the aerosols would make to the
aerosol radiative forcing (ΔF) per unit of aerosol optical depth (AOD) (Sheridan and Ogren,
1999) and we estimated the RFE at TOA as the Eq.6 (Haywood and Shine, 1995; Sheridan
and Ogren, 1999):
$RFE = \dfrac{\Delta F}{AOD} = -DS_0 T_{at}^2 (1-A_C) \times SSA \times \beta \times ((1-R_s)^2 - (\dfrac{2R_S}{\beta}) \times (\dfrac{1}{SSA}-1))$     (6)
where D is the fractional day length, $S_0$ is the solar constant, $T_{at}$ is the atmospheric
transmission, Ac is the fractional cloud amount, and Rs is the surface reflectance. The
constants used were D = 0.5, So = 1370 $Wm^{-2}$, $T_{at}$ = 0.76, Ac = 0.6, and Rs = 0.15 as
suggested by Haywood and Shine (1995), and upper scatter fraction β was calculated from
$\beta = 0.0817+1.8495 \times b - 2.9682 \times b^2$. backscatter fraction (b) was calculated based on
scattering coefficient ($\sigma_{sp}$) and backscattering coefficient ($\sigma_{bsp}$) measured by Nephelometer
as b = $\sigma_{bsp}/ \sigma_{sp}$. Equation (6) has been widely used to assess the intrinsic radiative forcing
efficiency of aerosols at the top of the atmosphere (Sheridan and Ogren, 1999; Virkkula et
al., 2011; Shen et al., 2018). Note that RFE in this study was in a dry condition. As the
backscatter fraction and single scattering albedo are all RH-dependent, the RFE is also
sensitive to RH (Fierz-Schmidhauser et al., 2010). Previous studies revealed that RFE
increased as the elevating RH (Titos et al., 2021; Xia et al., 2023). In this study, the values
of $\Delta F$ at TOA were also caculated by multiplying the RFE for $PM_{10}$ with the AOD of
ambient atmospheric aerosols observed at the CAMS site during the study periods. AOD
can be downloaded from Aerosol Robotic Network (AERONET). Note that RFE was at a
dry state, thus the $\Delta F$ at TOA here may be slightly underestimated.
**2.4. Other data used**
The hourly $PM_{2.5}$ and $PM_{10}$ mass concentrations were measured at Guan yuan station,
which is about 3km from the CAMS site. The data can be derived from the national air
quality real-time publishing platform (http://106.37.208.233:20035/). The hourly
meteorological data were measured at Haidian station (station No. 54399) and obtained
from the National Meteorological Information Center of China Meteorological
Administration.
**2.5. Back trajectories analysis**
To investigate the influence of air mass origins on aerosol optical properties, 48-h
backward trajectories arriving at Beijing at a height of 500 m above ground level were
calculated from 0:00 to 23:00 local time each day from March 2018 to February 2022,
using the Trajstat Software, combined with HYSPLIT 4 model (Hybrid Single-Particle
Lagrangian Integrated Trajectory), and the NCEP Global Data Assimilation System
(GDAS) data with a $1° \times 1°$ resolution (Draxler and Hess, 1998; Wang et al., 2009).

In this study, four seasons are defined as follows: spring from March to May, summer

from June to August, autumn from September to November, and winter from December to
the following February, and all data are reported in Beijing time (UTC+8).
**3 Results and discussion**
**3.1 Temporal variation of aerosol optical properties**

Figure 1 shows the annual variation of $\sigma_{ab}$, SSA, Rab, and $PM_{2.5}$ mass concentration

from 2018 to 2021. During the study period, the annual mean $PM_{2.5}$ in 2018 was 54.7 μg
$m^{-3}$, and it decreased by 34.4% (35.9 μg $m^{-3}$) in 2021, which suggested that the strict
pollution control measures are effective in reducing the PM loadings in Beijing (Lei et al.,
2021). Gong et al. (2022) demonstrated that emission reduction dominated the variations
of $PM_{2.5}$ mass concentration in Beijing from 2013 to 2020, and meteorology and emission
reduction contributed 7% and 63.2% of decreases, respectively. $\sigma_{ab}$ at 550 nm of $PM_{10}$ and
$PM_1$ showed similar annual variations. The annual mean $\sigma_{ab}$ at 550 nm of $PM_{10}$ and $PM_1$
decreased by 55.0% and 53.5%, respectively, from 2018 to 2021. The Mann–Kendall trend
test supported that the decrease in $\sigma_{ab}$ for $PM_1$ and $PM_{10}$ from 2018 to 2021 was significant
(Table S1). Carbonaceous aerosol, especially black carbon, is closely related to aerosol
absorption (Yang et al., 2009). A continuous decrease in $\sigma_{ab}$ was consistent with the
continuous reduction of black carbon concentration observed in Beijing in previous studies
(Ji et al., 2019; Sun et al., 2022), which was mainly related to significantly reduced primary
emissions caused by effective air pollution control measures in recent years (Xia et al.,
2020). The annual mean $\sigma_{ab}$ for $PM_{10}$ and $PM_1$ in 2021 was 9.8 Mm$^{-1}$ and 8.7 Mm$^{-1}$, which
were both lower than the result observed in Nainital, in the GH region, India (Dumka et al.,
2015), and the measurement at an urban site in Spain from March 2006 to February 2007
(Titos et al., 2012). In fact, with the emission reduction and improvement of air quality, the
aerosol scattering coefficient ($\sigma_{sp}$) for $PM_{10}$ and $PM_1$ also decreased in Beijing. Hu et al.
(2021) revealed that $\sigma_{sp}$ decreased by approximately 18.4% for $PM_{10}$, and 16.7% for $PM_1$
from 2018 to 2019 in Beijing. Atmospheric conditions also have an effect on aerosol optical
properties. The variations of meteorological parameters from 2018 to 2021 (Figure S4)
showed that pressure, wind speed, temperature, and RH varied slightly, while accumulated
precipitation increased in 2021 compared with the other 3 years. On the other hand, a
correlation analysis was made between aerosol optical properties and meteorological
parameters. The Pearson correlation coefficients (R) between $\sigma_{ab}$ and meteorological
parameters (Table S2) are lower than 0.5, indicating that a weak correlation (R<0.5) was
found between $\sigma_{ab}$ and meteorological parameters. This suggests that the meteorological
parameters' influence on $\sigma_{ab}$ is minor. Xia et al. (2020) revealed that the effect of emission
reduction was the major reason for the decrease of BC in Beijing. Actually, $\sigma_{ab}$ that was
observed at a background station in China and the European stations, which was with time
series longer than 10 years, also observed the reduction. $\sigma_{ab}$ showed a statistically
significant decreasing trend in Mt.Waliguan, a background station in China, from 2008–
2018 (Collaud Coen et al., 2020), which was similar to a decreasing trend of black carbon
(BC) in Mt.Waliguan from 2008-2017, mainly related to emission reduction (Dai et al.,
2021). A statistically significant decrease of 10-year $\sigma_{ap}$ was found in 12 stations in Europe,
which was similar to a decreasing trend in BC concentration in Europe related primarily to
traffic emission decreases (Collaud Coen et al., 2020).

SSA is a key variable in assessing the aerosol radiative forcing. The variation of SSA

also reflects the the ratio of aerosol scattering to total extinction with aerosol composition
changes. The annual variations of SSA for $PM_{10}$ and $PM_1$ were similar. During 2018-2021,
annual mean SSA at 550 nm increased from $0.89 \pm 0.04$ for $PM_{10}$ ($0.87 \pm 0.05$ for $PM_1$) in
2018 to $0.93 \pm 0.03$ for $PM_{10}$ ($0.91 \pm 0.04$ for $PM_1$) in 2021. Increasing SSA and decreasing
$PM_{2.5}$ mass concentration during the past four years suggested that the fraction of absorbing
aerosols became lower compared to scattering aerosols with the improvement of air quality
due to pollution control measure-taking. Collaud Coen et al. (2020) found that SSA
observed in Mt. Waliguan, a background station in Asia, presented an increasing trend
based on 10-year datasets, which were related to more recent abatement policies. The mean
submicron absorption ratio (Rab) increased yearly during the same period. It was from 86.1%
in 2018 to 89.2% in 2021, suggesting that fine particles are the main contributors to total
$PM_{10}$ absorption, and the contributions from fine particles to absorption became more
important.

The $\sigma_{ab}$, SSA, and AAE for $PM_1$ and $PM_{10}$ showed similar annual variations in all

seasons (Fig. 2 and Fig. S1). Thus, if not stated otherwise, the following discussion takes
the aerosol optical properties of $PM_{10}$ as an example. As shown in Fig. 2 seasonal average
of $\sigma_{ab}$ presented a continuous reduction during all seasons from 2018 to 2021, reflecting
the reduction of absorbing aerosols which were related to effective control of absorbing
aerosols emissions in Beijing. $\sigma_{ab}$ decreased by half in autumn and winter during the study
period, which was probably due to reducing coal consumption as a heating source and the
reduction of biomass burning. Compared with 2018, $\sigma_{ab}$ in the winter of 2019, 2020 and
2021 decreased by 3.0%, 24.9% and 53.2%, respectively. In the winter of 2019, the
lockdown of COVID-19 caused emission reduction from human activities in China (Le et
al., 2020; Tian et al., 2020), however, the unexpected smallest reduction of $\sigma_{ab}$ was
observed in the winter of 2019 compared with the winter of 2020 and 2021. This is related
to the fact that severe haze pollution still occurred in the North China Plain and BC
concentrations rose unexpectedly during the lockdown period (Liu et al., 2021; Jia et al.,
2021). In particular, $\sigma_{ab}$ for $PM_1$ and $PM_{10}$ decreased even up to 63% and 67% in the
summer from 2018 to 2021. Traffic is a relatively stable source of absorption aerosols in
summer (Li et al., 2022b). The largest deduction of $\sigma_{ab}$ was in summer and could be related
to more strict vehicle emission standards (Zhang et al., 2019).

In general, AAE was lowest in summer and highest in winter. The mean values of

AAE for $PM_{10}$ were 1.13 and 1.41 in summer and winter, respectively, similar to result at
an urban site in Beijing in 2018 (Xie et al., 2020). During summer, the average AAE was
generally close to 1, which suggested that BC from traffic emissions was the major
component of light-absorbing aerosols. Li et al. (2022b) found that the percentage of liquid
fuel (traffic) contributing to the total BC was 86.8% in summer in Beijing. The highest
AAE suggested that BrC contributed to light absorption strongest in winter, which is due
to enhanced emissions from biomass burning and coal combustion in winter (Sun et al.,
2018). Notably, AAE decreased in winter from 1.48 for $PM_{10}$ (1.48 for $PM_1$) in 2018 to
1.37 for $PM_{10}$ (1.34 for $PM_1$) in 2021 (Fig. 2 and Fig. S1), indicating a decreasing
contribution from BrC to light absorption, which may relate to the effect control of biomass
burning and coal combustion caused by changes in heating energy structure (Ji et al., 2022).
To improve air quality, the Beijing-Tianjin-Hebei region adjusted the energy structure
during the heating period and developed clean heating projects, such as the "coal to gas"
project (Zhao et al., 2020; Liu et al., 2019). During the whole period, AAE was similar in
spring and autumn indicating that light-absorbing aerosols were from similar emission
sources in spring and autumn (Ran et al., 2016). AAE slightly increased in spring and
autumn from 2018 to 2021. Part of the reason was the occurrence of multiple fugitive dust
in spring and autumn (Yi et al., 2021; Gui et al., 2022). On the other hand, BrC could also
be formed from secondary reactions (Bond et al., 2013; Wang et al., 2022). A slight
increase in AAE in spring and autumn may also have been caused by a greater amount of
secondary organic aerosol formation as a result of an increased atmospheric oxidation
capacity (Ji et al., 2019; Lei et al., 2021).
The seasonal mean SSA increased in all seasons from 2018 to 2021, indicating that
the contribution of scattering aerosols to extinction increased. This suggested that more
effective control of scattering aerosols should be attached more importance in order to
improve visibility in the future. In particular, SSA in winter increased significantly from
0.88 in 2018 to 0.93 in 2021, which revealed that the proportion of absorbing aerosols
decreases considerably in winter. This is consistent with recent research which suggests
that air pollution control measures has been more effective in reducing the primary
pollution emissions than secondary species (Vu et al., 2019; Sun et al., 2020). On the other
hand, seasonal mean SSA for $PM_{10}$ was 0.94±0.04, 0.94±0.04, 0.92± 0.04, 0.93±0.03 in
spring, summer, autumn, and winter 2021. Similar SSA suggests that the proportions of
light absorbing and scattering components became relatively stable in four seasons.

Figure 3 shows the diurnal variations of $\sigma_{ab}$ and SSA at 550 nm for $PM_{10}$, which are

similar to those for $PM_1$ (Figure. S2). In the past four years, $\sigma_{ab}$ was lower during the day
and higher at night in four seasons. This was consistent with that observed at an urban site
in Beijing during 2014-2016 (Wang et al., 2019). The evolution of the planetary boundary
layer had an important influence on the diurnal variation of the $\sigma_{ab}$. With stronger solar
radiation, the boundary layer was more fully developed during the daytime, and after sunset,
the convective boundary layer underwent a transition to the nocturnal stable boundary layer
(Guo et al., 2016). Furthermore, emissions also affected the diurnal variation of the $\sigma_{ab}$. For
example, heavy-duty diesel trucks and heavy-duty vehicles were only allowed to enter
urban areas from 23:00 to the following day 06:00 (Hu et al., 2021). As a response, the
minimum $\sigma_{ab}$ occurred during 12:00–18:00, when the planetary boundary layer was well-
developed, and truck emission was lower. With shallow boundary layer height and
enhanced emissions from heavy-duty trucks, $\sigma_{ab}$ reached the maximum at night. During the
study period, SSA showed a significant peak in the afternoon in four seasons, which was
similar to previous studies in urban Beijing (Zhao et al., 2019; Wang et al., 2019). Higher
SSA was shown in the afternoon, which was mainly related to the reduction of absorbing
aerosols emission, and more secondary scattering aerosol produced by strong chemical
reactions under intensive solar radiation and high temperature in the afternoon (Han et al.,

2017).

**3.2 Aerosol radiative effect**

To study the climate impact of the aerosol particles, we investigated the variation of

aerosol radiative forcing efficiency (RFE) at the top-of-the-atmosphere (TOA) variations.
As seen in Fig. 4, RFE for $PM_{10}$ and $PM_1$ were always negative during the whole
observation period, suggesting that the aerosols measured in urban Beijing have a stable
cooling effect on the climate. RFE for $PM_{10}$ and $PM_1$ at dry condition were $-27.0$ and -
$26.2$ W m$^{-2}$ AOD$^{-1}$ in 2021 in urban Beijing, which was slightly negative than that of $-24.9$
W m$^{-2}$ AOD$^{-1}$ in Nanjing (Shen et al., 2018) and highly negative than that of $-19.9$ W m$^{-2}$
AOD$^{-1}$ in Finland (Virkkula et al., 2011). This suggested that the aerosols in urban Beijing
have a higher cooling efficiency. In eq. (6) The fractional day length (D), solar constant
(So), atmospheric transmission ($T_{at}$), fractional cloud amount (Ac), and surface reflectance
(Rs) were constants, which were widely used in previous studies (Delene and Ogren, 2002;
Andrews et al., 2011; Sherman et al., 2015; Shen et al., 2018). These values are the globally
averaged values and don't always represent the conditions in Beijing, but using the same
constants makes it possible to compare the intrinsic forcing efficiency of the aerosols
measured at different stations around the world and to study how the RFE changes with
varying SSA and b (Sherman et al., 2015; Luoma et al., 2019). On the other hand, RFE is
sensitive to RH as the aerosol optical properties are different due to hygroscopic growth
(Fierz-Schmidhauser et al., 2010; Luoma et al., 2019). Previous studies demonstrate that
SSA increases with RH, while b decreases with increasing RH (Carrico et al., 2003; Cheng
et al., 2008). The change of SSA to increase with RH and of b to decrease with RH will
have opposite effects on the RFE, and thus to some extent, the RH dependencies of these
two parameters will counterbalance each other (Luoma et al., 2019). Titos et al. (2021)
found that the range of forcing enhancement in different types of sites varies from almost
no enhancement up to a factor of 3–4 at RH=90 %. The results observed in urban Beijing
showed that the aerosol radiative forcing at RH = 80 % was 1.48 times that under dry
conditions (Xia et al., 2023). RFE was calculated at a dry state in this study, while the
atmosphere is not generally dry in the ambient air. Thus, the RFE in this study does not
represent ambient conditions. The simplified RFE in this study does not represent the actual
value for the aerosol forcing; however, it can still indicate how the changes in aerosol
optical properties affect the climate (Delene and Ogren, 2002; Andrews et al., 2011;
Sherman et al., 2015). RFE was affected by SSA and backscatter fraction (b) and we
investigated the RFE variations with SSA and b in Beijing. As shown in Fig. 5, When SSA
increases from 0.7 to 0.92, the mean RFE increases by 1.59 times, suggesting that SSA
plays an important role in strengthening cooling efficiency. When SSA>0.92, the mean
RFE relatively keeps constant. The approximate constant RFE does not mean that the
absolute aerosol radiative forcing is constant; it just suggests that the intrinsic nature of the
aerosol will not significantly affect the calculation of RFE (Andrews et al., 2011). Also,
the backscatter fraction has a negative relationship with RFE. A lower values of backscatter
fraction corresponds to larger particles (Luoma et al., 2019). RFE became more negative
with increasing b, suggesting that smaller particles would cool the atmosphere more
efficiently. During the study period, SSA increased from 0.89 to 0.93, while the yearly
mean value of b was 0.13 every year during the study period. RFE became more negative
from 2018 to 2021, suggesting that the efficiency of the aerosol cooling atmosphere was
higher, which was mainly influenced by increasing SSA.
The ratio of $\Delta F/AOD$ is known as the aerosol radiative forcing efficiency (RFE) and
$\Delta F$ at TOA was caculated by multiplying the RFE for $PM_{10}$ with the AOD of ambient
atmospheric aerosols observed at the CAMS site during the study periods. The mean value
of $\Delta F$ from 2018-2021 was -15.0 W m$^{-2}$, -12.5 W m$^{-2}$, -12.1 W m$^{-2}$, and -11.8 W m$^{-2}$,
respectively. Although RFE became more negative, the annual mean $\Delta F$ in 2021
corresponding to lower columnar aerosol loading became less negative than that of 2018
corresponding to higher columnar aerosol loading (Fig. S3) which was consistent with the
analysis that aerosol loading was a essential factor for the estimation of $\Delta F$ (Andrews et al.,
2011; Delene and Ogren, 2002).
**3.3 Transport and its impact on aerosol optical properties in Beijing**

In addition to local emissions, regional transport is also an important source of

particulate matter in Beijing (Chang et al., 2019). Based on previous studies, aerosol source
regions and air mass pathways could also affect aerosol optical properties, and the different
origins of air masses showed different aerosol optical properties (Zhuang et al., 2015; Pu
et al., 2015). The air mass back-trajectories analysis in the North China Plain revealed that
the absorption coefficients and SSA were high when the air masses came from densely
populated and highly industrial areas (Yan et al., 2008). Therefore, air mass back-
trajectories were analyzed in this study to explore the regional transports' influence on
aerosol optical properties. First, the air mass back trajectories during 2018–2021 were
calculated and clustered (Fig. 7); then, we statistic the aerosol optical properties of each
cluster from 2018-2021 (Fig. 8). Based on the Euclidean distance, the back trajectories
were classified into five clusters, in which clusters 1, 2 and 3, which originated from the
clean areas in Mongolia and eastern Inner Mongolia, and transported to Beijing along the
pathway with low emissions, were corresponded to low $\sigma_{ab}$ and low $PM_{2.5}$ (Fig. 8a, d).
Cluster 4 from the south of Beijing and cluster 5 from the west of Beijing were referred to
as the polluted air masses, and the average $PM_{2.5}$ concentrations and $\sigma_{ab}$ of clusters 4 and
5 were higher than those of clusters 1, 2, and 3 in each year (Fig. 8a, d). Cluster 4 passed
through Shandong and Hebei Province, which was heavily polluted before arriving in
Beijing. Cluster 5 passed through polluted Shanxi and Hebei during transport. Higher $\sigma_{ab}$
and $PM_{2.5}$ mass concentrations were mainly distributed in clusters 4 and 5 each year.
Lower AAE in cluster 4 indicates that the southern air mass carries more freshly emitted
BC particles. SSA of cluster 4 from the south was higher (Fig. 8b), which may relate to
low $BC/PM_{2.5}$ ratios in south air masses (Xia et al., 2020). Zhang et al. (2013) found that
high levels of secondary inorganic aerosols related to high humidity were transported by
southern air masses, which enhanced heterogeneous reaction and leaded to relatively low
$BC/PM_{2.5}$ ratios. Fig. 7b showed percentage of each cluster accounting for the total back
trajectories in each year. The results indicated that variation in each cluster fraction from
2018 to 2021 was slight. In general, cluster 1-5 accounted for 19%-21%, 13%-17%, 16%-
20%, 29%-36%, 12%-20% of total back trajectories, respectively. Notably, the percentage
of polluted-relevant air masses (cluster 4 and cluster 5) was ~50% each year, indicating
that the transport from the south and the west of has a considerable impact on the aerosol
optical properties. $\sigma_{ab}$ corresponding to clusters 4 and 5 decreased by 47.3% and 58.4%,
and a decrease of $PM_{2.5}$ mass concentration from clusters 4 and 5 was 38.9% and 37.4%
during 2018 - 2021 (Fig. 8a, d), which may result from the air quality has improved caused
by control of source emissions in surrounding regions of Beijing. Therefore, the
comprehensive control of atmospheric pollution in Beijing and surrounding regions would
be highly effective in reducing air pollution in Beijing.

**4 Conclusions**

In this study, 4-year measurements of aerosol absorption properties and single scattering albedo for $PM_{10}$ and $PM_1$ in Beijing were analyzed. The annual mean $PM_{2.5}$ in 2018 was 54.7 μg m$^{-3}$, and it decreased by 34.4% (35.9 μg m$^{-3}$) in 2021, which suggested that the strict pollution control measures are effective in reducing the PM loadings in Beijing. The annual mean $\sigma_{ab}$ of $PM_{10}$ and $PM_1$ decreased by 55.0% and 53.5%, respectively, and it showed a similar decrease in all seasons. Significant reduction in $\sigma_{ab}$ may be related to reduced primary emissions caused by effective air pollution control measures. SSA at 550 nm increased from 0.89 ± 0.04 for $PM_{10}$ (0.87 ± 0.05 for $PM_1$) in 2018 to 0.93 ± 0.03 for $PM_{10}$ (0.91 ± 0.04 for $PM_1$) in 2021 and the seasonal averages of SSA for two sizes also increased in four seasons. Increasing SSA and decreasing $PM_{2.5}$ mass concentration suggest that the fraction of absorbing aerosols decreased with improved air quality due to pollution control measure-taking. During the study period, the annual average of Rab increased year by year and was up to 89.2% in 2021, indicating that fine particles are the main contributors to the total $PM_{10}$ particle absorption, and the contributions from fine particles to absorption became more important in Beijing.

During the study period, AAE was lowest in summer and highest in winter. Seasonal mean AAE in summer was generally close to 1 indicating that freshly emitted BC from traffic sources was a major component of light-absorbing aerosols. The highest AAE highlights the importance of BrC light absorption in winter. Notably, AAE in winter decreased from 2018 to 2021, implying a decreasing contribution from BrC to absorption, which may relate to the effective control of biomass burning and coal combustion caused

by changes in heating energy structure. AAE in spring and autumn was similar, indicating
light-absorbing aerosols were from similar emission sources in these two seasons.

Using a simple analytical equation, we investigated the aerosol radiative effect.

Aerosol radiative forcing efficiency (RFE) for $PM_{10}$ and $PM_1$ was always negative,
suggesting that the aerosols measured in urban Beijing have a stable cooling effect on the
climate. RFE for $PM_{10}$ and $PM_1$ at dry conditions were $-27.0$ and $-26.2$ $W\ m^{-2}\ AOD^{-1}$ in
2021 in urban Beijing. RFE was influenced by SSA and b. Higher b corresponds to more
negative RFE suggesting that smaller particles larger would cool the atmosphere more
efficiently. When SSA<0.92, the absolute value of mean RFE increased by 1.59 times,
suggesting that SSA plays an important role in strengthening cooling efficiency. When
SSA>0.92, the mean RFE keeps relatively constant, suggesting that the intrinsic nature of
the aerosol will not significantly affect the calculation of RFE. SSA increased from 0.89 to
0.93, while the yearly mean value of b was 0.13 every year during the study period. RFE
became more negative from 2018 to 2021, suggesting that the efficiency of the aerosol
cooling atmosphere was higher, which was mainly influenced by increasing SSA.

Regional transport and its impact on aerosol optical properties were also analyzed.

The air mass back trajectories arriving at Beijing were divided into five clusters. Clusters
1, 2, and 3, which originated from the clean area in Mongolia and eastern Inner Mongolia,
were transported to Beijing along the pathway with low emissions, corresponding to low
$\sigma_{ab}$ and low $PM_{2.5}$. Air masses from south and west (Cluster 4 and Cluster 5), which both
crossed the polluted region, always brought high $PM_{2.5}$ concentrations and $\sigma_{ab}$. $\sigma_{ab}$
corresponding to clusters 4 and 5 decreased by 47.3% and 58.4%, and a decrease of $PM_{2.5}$
mass concentration from clusters 4 and 5 was 38.9% and 37.4% during 2018 - 2021, which
may result from the control of source emissions in surrounding regions of Beijing.
Therefore, comprehensive control of atmospheric pollution in surrounding regions of
Beijing is conducive to reducing pollution in Beijing.
**Data availability.**
The data in this study are available at: https://doi.org/10.5281/zenodo.7466069 (Hu et
al., 2022)
**Competing interests.**
The authors declare that they have no conflict of interest.
**Author contributions.**
XH performed data analysis, prepared the figures and wrote the manuscript. JS
designed the experiment and outlined the manuscript. XH, CX, and JS conducted the
measurements. XS, YZ, QL, ZL, SZ, JW, AY, JL, SL and XZ discussed the results and
commented on the manuscript.
**Acknowledgments.**
This study was supported by the National Natural Science Foundation of China
(42090031, 41875147, 42075082, 42175128), Chinese Academy of Meteorological
Sciences (2022KJ002, 2022KJ005, 2020KJ001, 2020Z002). It was also supported by the
Innovation Team for Haze-fog Observation and Forecasts of MOST.

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

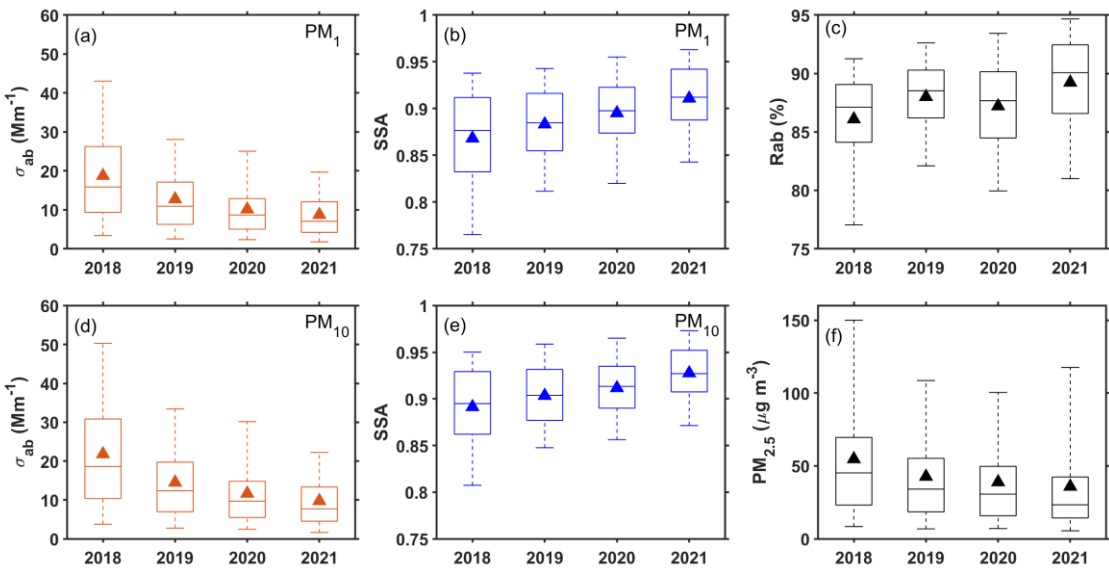


**Figure 1.** Annual variation of aerosol optical properties and PM₂.₅ mass concentration,
absorption coefficient $\sigma_{ab}$ at 550 nm for (a) PM₁ and (d) PM₁₀, SSA at 550 nm for (b) PM₁
and (e) PM₁₀, (c) Rab and (f)PM₂.₅ mass concentration. The solid line inside the box
represents the median and the triangle indicates the mean. The box contains the range of
values from 25% (bottom) to 75% (top), and the upper and lower whiskers are the 95th and
5th percentiles, respectively.

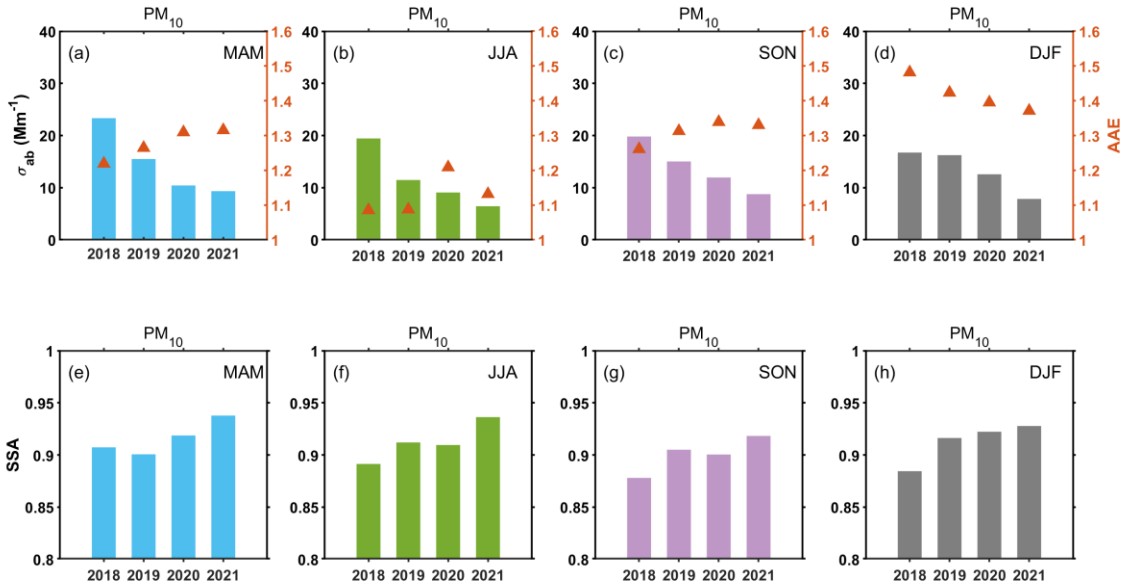


**Figure 2.** Seasonal variation of aerosol optical properties of $PM_{10}$ from 2018-2021, (a-d)
$\sigma_{ab}$ (bar) at 550 nm, $AAE_{450/700}$ (triangle), and (e-h) SSA (bar) at 550 nm.

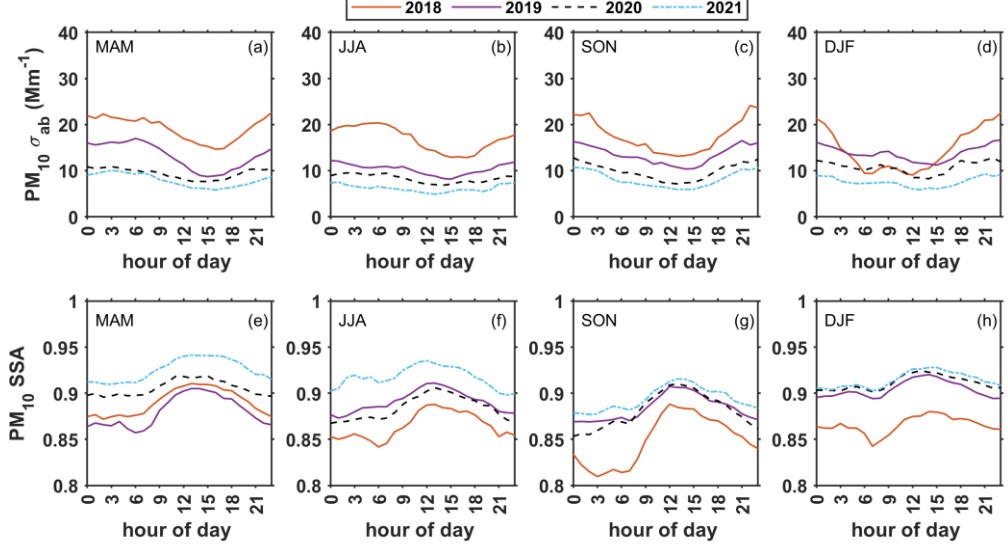


**Figure 3.** Diurnal variations of $\sigma_{ab}$ (a-d) and SSA (e-h) at 550 nm for $PM_{10}$ in four seasons
from 2018 to 2021.

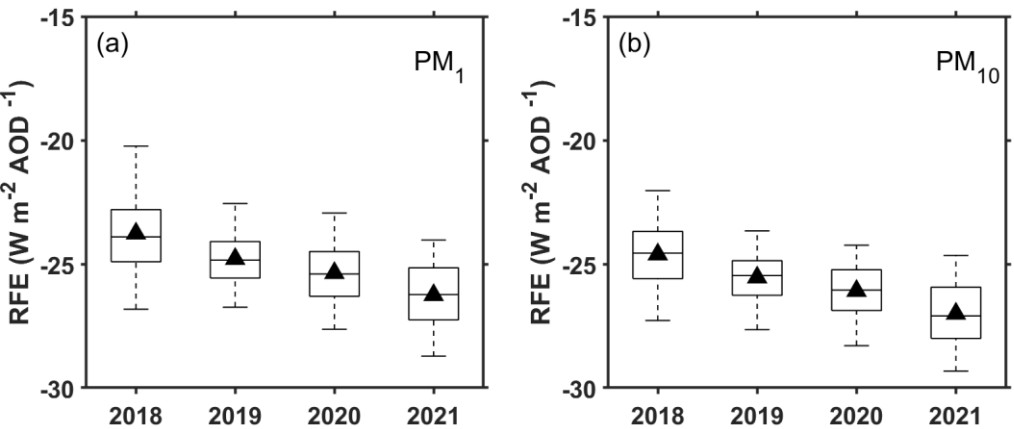


875 **Figure 4.** The annual variation of aerosol radiative forcing efficiency for $PM_1$ (a) and $PM_{10}$

876 (b). The solid line inside the box represents the median, and the triangle indicates the mean.

877 The box contains the range of values from 25% (bottom) to 75% (top), and the 95th and

878 5th percentiles, respectively.


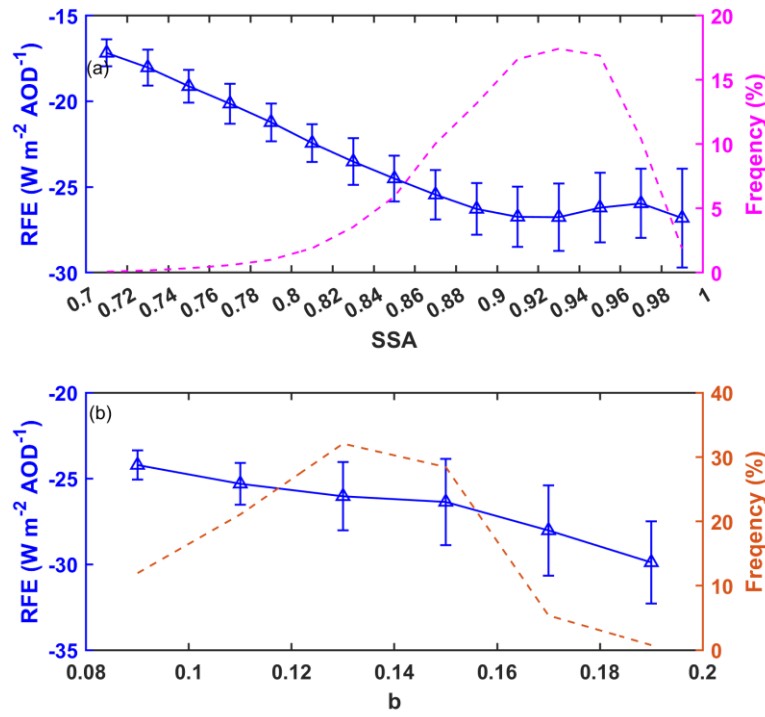


**Figure 5.** The relationship of RFE with (a) SSA and (b) backscatter fraction. The pink dash
line represents the frequency distribution of SSA (a) and the brown dash line represents the
frequency distribution of backscatter fraction (b).

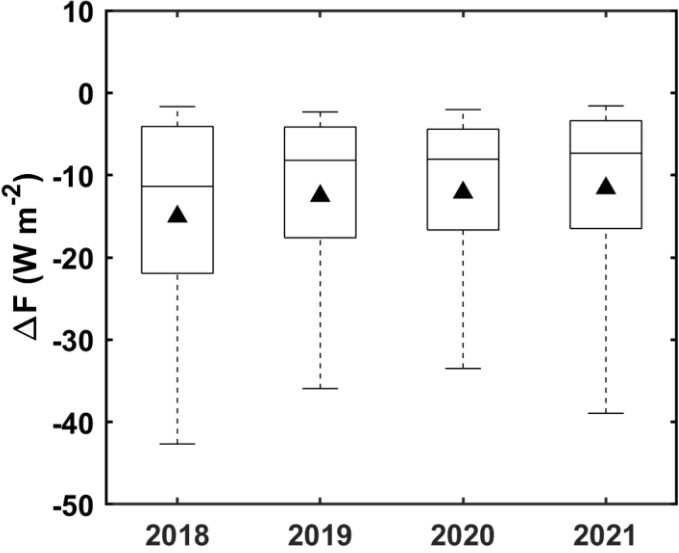


**Figure 6.** Annual variation of aerosol radiative forcing (ΔF) at TOA from 2018 to 2021
calculated from daily mean data. The solid line inside the box represents the median, and
the triangle indicates the mean. The box contains the range of values from 25% (bottom)
to 75% (top), and the 95th and 5th percentiles, respectively.

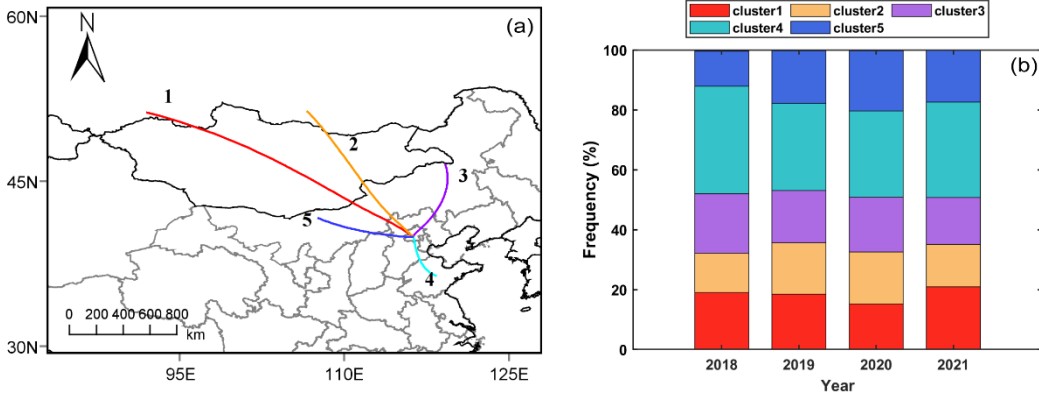


**Figure 7.** (a) Air mass clusters of back trajectories arriving in Beijing during 2018–2021
and (b) the fraction of each cluster accounting for the total back trajectories in each year.

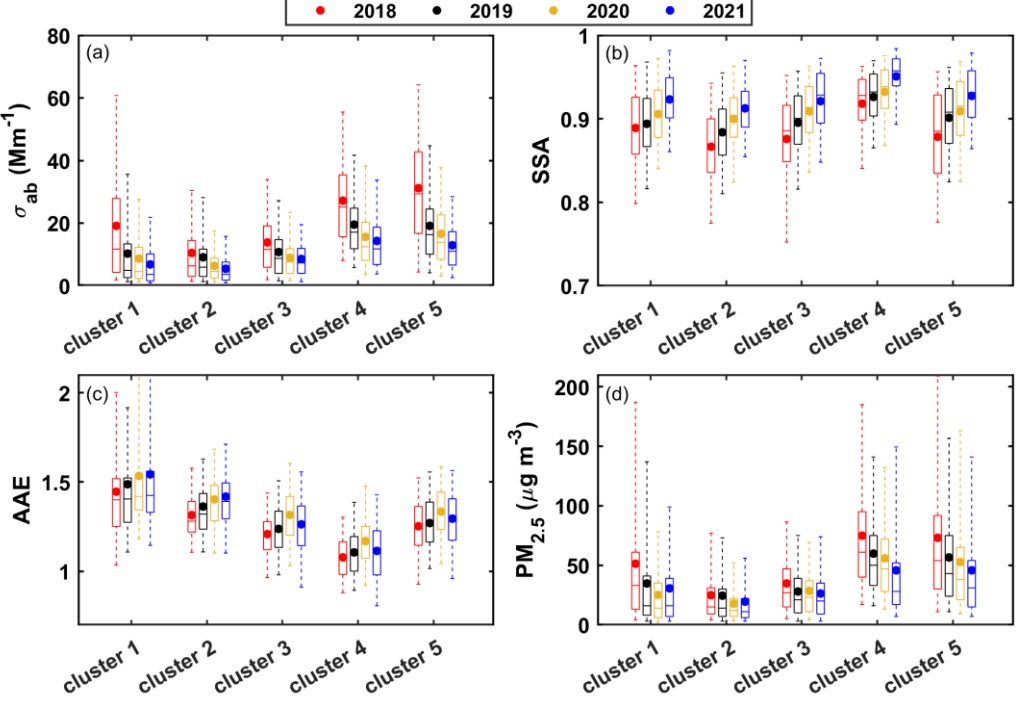


**Figure 8.** The variation of (a) $\sigma_{ab}$, (b) SSA, (c) AAE, and PM$_{2.5}$ mass concentration in each
cluster from 2018 to 2021. The solid line inside the box represents the median and the dot
indicates the mean. The box contains the range of values from 25% (bottom) to 75% (top),
and the 95th and 5th percentiles, respectively.