# Peer review of "Measurement Report: Rapid decline of aerosol absorption coefficient and aerosol optical properties effects on radiative forcing in an urban area of Beijing from 2018 to 2021"

_Atmospheric Chemistry and Physics, 2022_

## Author Comment (AC1)

**Response to Reviewers' comments**

We are thankful to the two reviewers for their thoughtful and constructive comments that help us improve the manuscript substantially. We have revised the manuscript accordingly. Listed below is our point-to-point response in blue to each comment that was offered by the reviewers.

**Response to Reviewer #1**

General comments:

The authors presented 4-year optical properties of atmospheric aerosols collected over two size fractions (PM1 and PM10). The analysis includes scattering and absorbing coefficients, along with derived values of SSA, AAE, and RFE. The manuscript is submitted as a 'Measurement Report". Four-year continuous datasets from a highly populated city like Beijing, China, is beneficial.

**R:** The authors thank the reviewer's positive comments.

**Major comments:**

According to the authors, the policies implemented to improve the air quality in Beijing are the main reason for the reduction in optical properties magnitude from 2018-2021. The policies effectively reduced the absorbing aerosols, which reduced the absorption coefficients. Does the same reduction happen in the scattering coefficient also? The absorbing aerosols like black carbon, BrC, and Dust also scatter in nature. So, for a complete picture, it will be better to see the fate of scattering coefficients over the years.

**R:** Thank you for the reviewer's advice. Since the two-year scattering coefficients has been published (Hu et al., 2021), therefore, this manuscript focused on optical properties other than scattering coefficients. The variation of aerosol scattering properties for submicron ($PM_1$) and sub-10 μm particles ($PM_{10}$) under dry conditions (RH <30%) in Beijing from 2018 to 2019 has been analyzed in our previous study (Hu et al., 2021). The results showed that $\sigma_{sp}$ for $PM_1$ and $PM_{10}$ showed the similar variation. $\sigma_{sp}$ in 2019 decreased by approximately 18.4% for $PM_{10}$, and 16.7% for $PM_1$ compared with those in 2018. In addition, aerosol scattering coefficient is highly positively correlated with PM mass concentration. Decreasing $PM_{2.5}$ mass concentrations from 2018-2021 may reflect the variation of aerosol scattering coefficient to some extent. Based on our observed data, the annual mean of $\sigma_{sp}$ was 160.4 Mm$^{-1}$ in 2018, and it decreased by 19.6% (128.9 Mm$^{-1}$) in 2021, shown in the following Figure R1 which does not include in the revised manuscript.

[Figure]

**Figure R1.** Annual variation of scattering coefficient ($\sigma_{sp}$) at 550 nm for (a) PM$_1$ and (b) PM$_{10}$.

We cited the results of aerosol scattering coefficient in the previous study (Hu et al., 2021), and added the sentences in the revised manuscript as follow:
"In fact, with the emission reduction and improvement of air quality, the aerosol scattering coefficient ($\sigma_{sp}$) for PM$_{10}$ and PM$_1$ also decreased in Beijing. Hu et al. (2021) revealed that $\sigma_{sp}$ decreased by approximately 18.4% for PM$_{10}$, and 16.7% for PM$_1$ from 2018 to 2019 in Beijing." (Line 245-248 in revised manuscript)

**Reference:**
Hu, X., Sun, J., Xia, C., Shen, X., Zhang, Y., Zhang, X., and Zhang, S.: Simultaneous measurements of PM1 and PM10 aerosol scattering properties and their relationships in urban Beijing: A two-year observation, Sci. Total Environ., 770, 145215, 10.1016/j.scitotenv.2021.145215, 2021.

Figure 1 shows the overall reduction in PM$_{2.5}$. So, it is safe to assume that the PM$_1$ and PM$_{10}$ mass concentrations were also proportionally reduced. The general decrease in PM will automatically lessen the optical properties irrespective of the type of aerosol controlled.
**R:** We agree the reviewer's opinion. We did not express the meaning clearly in the original manuscript, and revised as the following:
"Carbonaceous aerosol, especially black carbon, is closely related to aerosol absorption (Yang et al., 2009). A continuous decrease in $\sigma_{ab}$ was consistent with the continuous reduction of black carbon concentration observed in Beijing in previous studies (Ji et al., 2019; Sun et al., 2022), which was mainly related to significantly reduced primary emissions caused by effective air pollution control measures in recent years (Xia et al., 2020)." (Line 237-242 in revised manuscript)
**Reference:**
Yang, M., Howell, S. G., Zhuang, J., and Huebert, B. J.: Attribution of aerosol light absorption to black carbon, brown carbon, and dust in China – interpretations of atmospheric measurements during EAST-AIRE, Atmos. Chem. Phys., 9, 2035–2050, 2009.

Ji, D., Gao, W., Maenhaut, W., He, J., Wang, Z., Li, J., Du, W., Wang, L., Sun, Y., Xin, J., Hu, B., and Wang, Y.: Impact of air pollution control measures and regional transport on carbonaceous aerosols in fine particulate matter in urban Beijing, China: insights gained from long-term measurement, Atmos. Chem. Phys., 19, 8569-8590, 10.5194/acp-19-8569-2019, 2019.

Sun, J., Wang, Z., Zhou, W., Xie, C., Wu, C., Chen, C., Han, T., Wang, Q., Li, Z., Li, J., Fu, P., Wang, Z., and Sun, Y.: Measurement report: Long-term changes in black carbon and aerosol optical properties from 2012 to 2020 in Beijing, China, Atmos. Chem. Phys., 22, 561-575, 10.5194/acp-22-561-2022, 2022.

Xia, Y., Wu, Y., Huang, R. J., Xia, X., Tang, J., Wang, M., Li, J., Wang, C., Zhou, C., and Zhang, R.: Variation in black carbon concentration and aerosol optical properties in Beijing: Role of emission control and meteorological transport variability, Chemosphere, 254, 126849, 10.1016/j.chemosphere.2020.126849, 2020.

**Minor comments:**

Line 88-89 – The term $\sigma_{ap}$ is not defined.
**R:** Corrected.

Figure 5 – Use a different color for the backscattering ratio (b) frequency line.
**R:** Changed as suggestion.

The author contribution list is incomplete.
R: We completed the authors' contribution list in the revised manuscript.

**Response to Reviewer #2**
Review of "Measurement Report: Rapid decline of aerosol absorption coefficient and aerosol optical properties effects on radiative forcing in urban areas of Beijing from 2018 to 2021" by Xinyao Hu and coauthors.

This manuscript presents 4 years of aerosol optical properties in $PM_{10}$ and $PM_1$ size fractions at an urban site in Beijing. The dataset is interesting, and the manuscript is well written. I'm not sure if it is completely appropriate for ACP since the scope of the results is quite limited.

R: The authors thank reviewer's comments. Providing reliable observations of aerosol optical properties are crucial for quantifying the aerosol radiative forcing on climate. The 4-year data presented in this study provide critical optical parameters for radiative forcing assessment within two size ranges and are also helpful for evaluating the effectiveness of clean air action. We revised the manuscript carefully according to reviewers' suggestions and comments, hopefully, it is suitable to the scope of the ACP.

General comment: The authors have to make clear that talking about trends require long-term data, and apply statistical tests to verify if the decrease observed is statistically significant or not (most likely it is, but need verification). Another concern is related to atmospheric conditions. Changes in meteorological conditions over time,

as well as the COVID-19 restrictions could be behind the observed "trend". However, none of these causes are discussed in the manuscript. There have been many publications about the decrease of pollutants in 2020 during COVID-19 lockdown, but this is not even mentioned in this manuscript. Furthermore, meteorological variables such as temperature, wind, precipitation, should be proven to be similar from year to year, so the results can be more conclusive.

R: Thanks for the reviewer's suggestion. The trends of the monthly $\sigma_{ab}$ was analyzed using the Mann–Kendall (MK) method at a 95% confidence level, with the statistical parameters listed in Table S1. The Mann–Kendall trend test supported that the decrease in $\sigma_{ab}$ from 2018 to 2021 was significant. We agree that at least 10 years of data are needed for climatic trend analysis. 4-year data presented in the manuscript can't represent the aerosol optical properties climatic trend. We just discuss the variation of $\sigma_{ab}$ in 4 years in the revised manuscript as the following.:

"The annual mean $\sigma_{ab}$ at 550 nm of $PM_{10}$ and $PM_1$ decreased by 55.0% and 53.5%, respectively, from 2018 to 2021. The Mann–Kendall trend test of monthly mean $\sigma_{ab}$ for $PM_1$ and $PM_{10}$ supported that the decrease in $\sigma_{ab}$ for $PM_1$ and $PM_{10}$ from 2018 to 2021 was significant (Table S1)." (Line 234-237 in revised manuscript)

Table. S1 Mann–Kendall trend test results (p = 0.05) for monthly mean value of $\sigma_{ab}$ for $PM_1$ and $PM_{10}$ from 2018 to 2021. Z is the standardized test statistic value.

| | Trend | Z | sen's slope |
|---|---|---|---|
| $PM_1\ \sigma_{ab}$ | Decreasing trend | -5.8134 | -0.24856 |
| $PM_{10}\ \sigma_{ab}$ | Decreasing trend | -5.643 | -0.29905 |

As the reviewer's suggestions, atmospheric conditions also have the effect on aerosol optical properties. So, we calculated the annual mean of meteorological parameters from 2018 to 2021. The results show that pressure, wind speed, temperature, RH varied slightly, while accumulated precipitation increased in 2021 compared with other 3 years. On the other hand, correlation analysis was made between aerosol optical properties and meteorological parameters. The Pearson correlation coefficients (R) between $\sigma_{ab}$ and meteorological parameters are lower 0.5, indicating that weak correlation (R<0.5) was found between $\sigma_{ab}$ and meteorological parameters. This suggests that the meteorological parameters influence on aerosol optical properties is minor, which is confirmed by previous study (Gong et al., 2022).

We added the discussion in the revised manuscript as follow:

"Gong et al. (2022) demonstrated that the emission reduction dominated the variations of $PM_{2.5}$ mass concentration in Beijing from 2013 to 2020, and meteorology and emission reduction contributed 7% and 63.2% of decreases, respectively." (Line 231-233 in revised manuscript)

"Atmospheric conditions also have an effect on aerosol optical properties. The variations of meteorological parameters from 2018 to 2021 (Figure S4) showed that pressure, wind speed, temperature, and RH varied slightly, while accumulated precipitation increased in 2021 compared with the other 3 years. On the other hand, a correlation analysis was made between aerosol optical properties and meteorological parameters. The Pearson correlation coefficients (R) between $\sigma_{ab}$ and meteorological parameters (Table S2) are lower than 0.5, indicating that a weak correlation (R<0.5) was found between $\sigma_{ab}$ and meteorological parameters. This suggests that the meteorological parameters' influence on $\sigma_{ab}$ is minor. Xia et al. (2020) revealed that the effect of emission reduction was the major reason for the decrease of BC in Beijing." (Line 248-257 in revised manuscript)

Table S2. Pearson correlation coefficient (R) between different aerosol optical properties and meteorological parameters (* Significant at $p < 0.05$).

|  | pressure | temperature | RH | precipitation | wind speed |
|---|---|---|---|---|---|
| PM$_1$ $\sigma_{ab}$ | -0.10* | -0.05* | 0.30* | -0.03* | -0.31* |
| PM$_{10}$ $\sigma_{ab}$ | -0.10* | -0.05* | 0.30* | -0.03* | -0.30* |

[Figure]

Figure S4. The annual variation of meteorological parameters from 2018 to 2021.

As the mentioned by the reviewer, our study period covered the period of the outbreak of COVID-19, which produced serious impacts in China. We added the discussion of the influence of COVID-19 in the revised manuscript.

"Compared with 2018, $\sigma_{ab}$ in the winter of 2019, 2020 and 2021 decreased by 3.0%, 24.9% and 53.2%, respectively. In the winter of 2019, the lockdown of COVID-19 caused emission reduction from human activities in China (Tian et al., 2020; Le et al., 2020), however, the unexpected smallest reduction of $\sigma_{ab}$ was observed in the winter of 2019 compared with the winter of 2020 and 2021. This is related to the fact that severe haze pollution still occurred in the North China Plain and BC concentrations rose unexpectedly during the lockdown period (Liu et al., 2021; Jia et al., 2021)." (Line 287-294 in revised manuscript)

|  | Trend | Z | sen's slope |
|---|---|---|---|
| $PM_1$ $\sigma_{ab}$ | Decreasing trend | -5.8134 | -0.24856 |
| $PM_{10}$ $\sigma_{ab}$ | Decreasing trend | -5.643 | -0.29905 |

L293: There are many assumptions behind the calculation of RFE and the impact of this assumptions should be discussed in more detail. The RH impact in RFE should be included here, so the results are relative to ambient conditions and not dry conditions.

R: Thanks for reviewer's suggestion. We added the discussion about the impact of assumptions behind the calculation of RFE including RH impact. The modified sentence as follow:

[revised manuscript text omitted]

Section of cluster analysis: not sure if this section is very informative relative to the main objective of the paper. Further discussion on the changes observed over the years is difficult to draw.

R: Thanks for this comment. Based on cluster analysis, we try to give some information on how the aerosol optical properties vary with the air mass pathways. We revised this section carefully and found we miss-matched the data when we plotted Figure 8. So, we corrected the Figure 8 and revised the paragraph as follow:

[revised manuscript text omitted]